# Nematicidal Efficacy of a dsRNA-Chitosan Formulation Against *Acrobeloides nanus* Estimated by a Soil Drenching Application

**DOI:** 10.3390/biology14091161

**Published:** 2025-09-01

**Authors:** Taegeun Song, Falguni Khan, Yonggyun Kim

**Affiliations:** School of Life Sciences & Engineering, Gyeongkuk National University, Andong 36720, Republic of Korea; xorms5095@naver.com (T.S.); falguni.agri@gmail.com (F.K.)

**Keywords:** RNA interference, dsRNA, chitosan, nematode, *Acrobeloides nanus*

## Abstract

A novel pesticide based on sprayed double-stranded RNA (dsRNA) has recently been developed to control insect pests, and its application is expected to expand for controlling various other pests. This study demonstrates its application to control a soil nematode, *Acrobeloides nanus*, by a drenching method. In this study, a bioassay used a soil sample containing the nematodes, which was treated with a formulated dsRNA with chitosan. Among the three target genes, *vATPase subunit B* was the most potent for killing the nematodes when its expression was specifically suppressed by the gene-specific dsRNA treatment. In this treatment, the chitosan formulation significantly enhanced dsRNA stability under soil conditions. These findings suggest a promising approach using chitosan-formulated dsRNA to control various soil-dwelling nematodes that damage crops.

## 1. Introduction

RNA interference (RNAi) is a cellular process to control gene expression at a posttranscriptional level [1]. Its gene-specific control is accomplished by a complementary short-interference single-stranded RNA produced by a specific RNase called Argonaute (Ago) in an RNA-induced silencing complex (RISC) [2]. Even though it is an endogenous process that regulates development, it is useful to defend against viral infections in plants and animals [3]. Based on this background, the exogenous application of double-stranded RNA (dsRNA) has been devised, and its successful manipulation of specific target genes in various fields including pest control in agriculture has been demonstrated [4]. Recently, the development of a spray-induced gene silencing (SIGS) technique has allowed us to practically apply dsRNAs to control various insect and plant pests [5].

Nematodes are a protostome invertebrate phylum and are classified into 12 different clades [6]. De Ley and Blaxter [7] grouped the clades into two classes, i.e., Enoplea (clades 1 and 2) and Chromadorea (clades 3 to 12), based on molecular and morphological criteria. Thus, a model nematode, *Caenorhabditis elegans*, is classified into clade 9A in Chromadorea. RNAi has been described as systemic in *C. elegans*, and it can also be transmitted to the next generation (parental RNAi) upon dsRNA injection or feeding [8]. This suggests that RNAi can be a novel tactic to control nematode pests in agriculture.

Root-knot nematodes cause significant crop losses by forming root galls to prevent nitrogen fixation, which may lead to total failure in yield [9]. Pioneering works to apply RNAi to control the nematode pests were performed using transgenic plants expressing dsRNA. For instance, a previous observation [10] demonstrated that the transgenic strains of *Arabidopsis thaliana* expressed against the dsRNA specific to the 16D10 gene, which is responsible for nematode–host integration, resulted in the reduced infectivity of four species of *Meloidogyne* (*M. incognita*, *M. javanica*, *M. arenaria*, and *M. hapla*). A similar host-induced gene silencing (HIGS) event occurred in another crop, *Vitis* sp. (grape), and reduced the infection of *M. incognita* [11]. The use of RNAi to control parasitic root nematodes was also effective for other targets in *Meloidogyne* such as FMRFamide-like peptide (FLP) genes responsible for modulating nerve and muscle activities [12], as well as for other genera such as *Pratylenchus* and *Heterodera* [13,14].

To avoid any concerns regarding genetically modified organisms, SIGS would be an ideal strategy to control the nematode pests. SIGS can effectively deliver dsRNA to target pests using nanoformulations such as chitosan or BioClay. However, few studies have evaluated SIGS, probably due to a lack of suitable dsRNA formulations against soil nematodes. For example, SIGS-based nematode control was devised through dsRNA formulations using a kaolinite nanoclay, in which *Globodera pallida*, a white potato cyst nematode and a major economic pest that causes substantial potato yield losses, is the target nematode [15]. dsRNAs specific to three FLP genes were formulated using the kaolinite nanoclay. The drenching of potato with the dsRNA–kaolinite formulations induced the deformation and premature death of J2 juveniles, compared with untreated J2s that entered J3 or J4 stages.

This study aimed to test a SIGS-based strategy for controlling soil nematodes. To this end, we used a soil nematode, *Acrobeloides nanus*, because of its ease of collection and relatively rapid reproduction in laboratory conditions [16,17]. After screening the optimal target genes using the unformulated dsRNAs, this study demonstrated the efficacy of the chitosan-based formulation on the nematicidal activity in the soil conditions.

## 2. Materials and Methods

### 2.1. Nematode Collection and Rearing

Nematode specimens were isolated from soil samples collected from a cucumber (*Cucumis sativus* L.) farm in Gunwi, Korea (36°11′32″ N, 128°33′51″ E) using a combination of sieving and the Baermann funnel method [18]. The isolated nematodes were then transferred to 1% agar plate containing a bacterial lawn of *Pseudomonas fluorescens* ANU101. The plates were incubated under darkness at 25 °C to facilitate nematode growth.

### 2.2. Nematode Identification Using Morphological Characteristics

Adult females were fixed in warm water (~60 °C) in 4% formaldehyde solution. Following fixation, they underwent dehydration using Seinhorst’s method [19] and were subsequently mounted in pure glycerin on slides [20]. A glass rod was used to prevent the samples from flattening while the coverslip was overlaid. Morphological characteristics were examined using a stereoscopic microscope (Stemi SV11, Zeiss, Jena, Germany) and a light microscope (Eclipse 80i, Nikon, Tokyo, Japan) at 50× magnification.

### 2.3. Chemicals Common in Different Treatments

Diethyl pyrocarbonate (DEPC)-treated water was prepared by adding DEPC to the deionized distilled water to 0.1% and autoclaved. Metafectene PRO was obtained from Biontex (München, Germany) to create liposomes following its addition to dsRNA. Proteinase K (20 µg/mL) was obtained from Biofact (Daejeon, Republic of Korea). Dithiothreitol (DTT), Tween-20, yeast tRNA, and N-2-hydroxyethylpiperazine-N-2-ethane sulfonic acid (HEPES) were purchased from Sigma-Aldrich Korea (Seoul, Republic of Korea). Methanol was obtained from Samchun Chemical (Seoul, Republic of Korea). Formamide, sodium citrate, sodium chloride, heparin, acetic anhydride, triethanolamine, dimethyl sulfoxide (DMSO), formaldehyde, and sodium dodecyl sulfate were purchased from Thermo Fisher Scientific (Waltham, MA, USA). Phosphate-buffered saline (PBS) was prepared with 0.1 M phosphate and 0.7% NaCl and its pH was adjusted to 7.4.

### 2.4. Nematode Identification Using Molecular Marker

For sequencing internal transcribed spacer (ITS) to identify the nematode isolate, genomic DNA (gDNA) was extracted from 500–600 nematodes which were collected from the laboratory population. Briefly, nematodes were crushed with a pestle in 50 μL of DNA extraction solution (LGC Bioresearch Technologies, Hoddesdon, UK) and heated at 100 °C for 10 min. After cooling on ice for 2 min, the suspension was centrifuged at 14,000× *g* for 5 min. The resulting supernatant (1–2 ng DNA/μL) was used as the gDNA sample.

Primers to amplify the ITS region were 5′-TCCGTAGGTGAACCTGCGG-3′ and 5′-TCCTCCGCTTATTGATATGC-3′ [21]. Each 25 μL PCR reaction consisted of 3 μL of gDNA, 2 mM MgCl_2_, 0.2 mM dNTP, 4 pmol of each primer, and 1 unit of Taq DNA polymerase (GeneAll, Seoul, Republic of Korea). PCR was performed on a My Cycler Personal Thermal Cycler (Bio-Rad, Hercules, CA, USA) using the following cycling conditions: initial heating at 94 °C for 3 min, 35 cycles of denaturation at 94 °C for 1 min, annealing at 48 °C for 1 min, and polymerization at 72 °C for 10 min, followed by keeping at 4 °C. PCR products were cloned into PCR2.1 cloning vector (Invitrogen, Carlsbad, CA, USA) and transformed into *Escherichia coli* TOP 10 chemically competent cells. Plasmids were obtained after cloning and used for bidirectional sequence analyses using M13F and M13R universal primers. Sequencing was performed by Macrogen (Seoul, Republic of Korea).

### 2.5. Prediction of Three Target Genes from the Nematode Genome

To screen the optimal target genes for developing dsRNA pesticide against *A. nanus*, Profilin/Actin-binding protein-10 (*Pat-10*), vacuolar type ATPase subunit B (*vATPase-B*) of *C. elegans*, and Uncoordinated-87 (Unc-87) of *Pratylenchus goodeyi* were used to obtain the orthologous sequences from *A. nanus* genome using BlastN (https://www.ncbi.nlm.nih.gov/, accessed on 1 July 2025). To obtain the open reading frame (ORF) of the collected sequences, the ORF finder tool from NCBI was used (https://www.ncbi.nlm.nih.gov/orffinder/, accessed on 1 July 2025). Prediction of the protein domain structure was performed using EMBL-EBI (www.ebi.ac.uk, accessed on 1 July 2025) and Pfam (http://pfam.xfam.org, accessed on 1 July 2025). Phylogenetic analysis was performed using the Neighbor-Joining method and the Poisson correction model using MEGA6.06 software (www.megasoftware.net, accessed on 1 July 2025). Bootstrapping values were obtained with 1000 replications to test the supports on each node in the resulting phylogenetic tree.

### 2.6. RNA Extraction, cDNA Preparation, and RT-PCR

Total RNA was extracted from approximately ~500–600 nematodes using Trizol reagent (Invitrogen, Carlsbad, CA, USA) according to the manufacturer’s instructions. The extracted RNA was resuspended in nuclease-free water and its concentration was measured with a spectrophotometer (NanoDrop, Thermo Scientific, Wilmington, DE, USA). For cDNA synthesis, 65 ng of RNA per reaction was used with RT-Premix (Intron Biotechnology, Seongnam, Republic of Korea) containing an oligo(dT) primer, following the manufacturer’s protocol.

For RT-PCR, the synthesized cDNAs were used for PCR amplification with Taq DNA polymerase and gene-specific primers under the following conditions: an initial denaturation at 95 °C for 3 min, followed by 35 cycles of denaturation at 95 °C for 1 min, different annealing temperatures (Appendix A) for 1 min, and extension at 72 °C for 1 min. The PCR reaction mixture (25 µL) consisted of a DNA template, dNTPs (2.5 mM each), 10 pmol of each primer, and Taq polymerase (2.5 units/µL).

Quantitative PCR (qPCR) was performed using a real-time PCR instrument (Step One Plus Real-Time PCR System, Applied Biosystems, Singapore) and Power SYBR Green PCR Master Mix (Life Technologies, Carlsbad, CA, USA), following the guidelines of Bustin et al. [22]. The qPCR reaction mixture (20 µL) included 10 µL of Power SYBR Green PCR Mix, 2 µL of cDNA template (65 ng/µL), and 1 µL each of forward and reverse primers (Appendix A). Elongation factor 1 alpha (*EF1α*) was used as the reference gene with its specific primers listed in Appendix A. Melting curve analysis was performed to verify the presence of a single specific PCR product. Quantitative analysis was conducted using the comparative CT (2^−∆∆CT^) method [23] with three independent replicates for each experiment.

### 2.7. dsRNA Preparation

Template DNA was amplified by PCR using gene-specific primers (Appendix A) with the T7 promoter sequence attached to their 5′ ends. The manufacturer’s instructions subjected the PCR products to in vitro transcription using the MEGAscript RNAi Kit (Ambion, Austin, TX, USA). The synthesized dsRNAs were mixed with Metafectene PRO at a 1:1 volume ratio and incubated at room temperature for 30 min to create a dsRNA–liposome complex. RNAi efficiency was evaluated via RT-qPCR 72 h after treatment.

### 2.8. Bioassay to Test Gene Silencing by dsRNA Application in an Agar Plate Assay

For the bioassay, 1% agar solution was prepared by dissolving 1 g of agar in 100 mL of distilled water and sterilizing it in an autoclave. The agar solution (3 mL) was dispensed to each well of a 24-well plate. dsRNA solutions were prepared at final concentrations of 50, 100, and 500 µg/µL by dilution with distilled water. A 20 µL volume of each dsRNA solution, including the control dsRNA (dsCON), was added to the designated well. Nematodes were suspended in 100 µL of PBS on a slide glass, and 40 individuals were transferred into each well using a 10 µL pipette. The plate was sealed with a lid and sealing tape and wrapped in aluminum foil to prevent light exposure. After three days at 25 °C, live nematodes were counted under a stereomicroscope by ‘S’ body shape and movement. Each treatment was replicated three times.

### 2.9. Bioassay to Test Gene Silencing by dsRNA Application in Soil

Instead of agar, this assay used soil (1.13 g/well). The sterilized soil was mixed with dsRNA suspension (100 µL) at 50, 100, and 500 µg/g. Control soil was mixed with PBS. A total of 100 nematodes were added to each well in PBS. The plates were then incubated in darkness at room temperature for seven days. Then the live nematodes in each well were counted using the Baermann funnel method. Each treatment was replicated three times.

### 2.10. Whole-Mount In Situ Hybridization of vATPase-B

To investigate the mRNA expression of *vATPase-B* in the juvenile stage of *A. nanus*, a whole-mount in situ hybridization was performed according to the method of Wang et al. [24]. After dsRNA feeding, nematodes were incubated in 1 mL of a lysis buffer (10 mM DTT, 0.1% Tween-20 in DEPC-treated water, pH 9.0) for 15 min at room temperature and then washed three times with ice-cold PBS. Samples were digested with proteinase K (20 µg/mL; Sigma-Aldrich Korea) for 10–15 min at room temperature and post-fixed in 3.7% formaldehyde (Sigma-Aldrich Korea) for 10 min, followed by additional PBS washes. Fixation was carried out in pre-chilled Dent’s fixative (methanol: DMSO, 8:2, *v*/*v*) for 20 min on ice. Samples were rehydrated in 50% methanol in PBS for 10 min and then post-fixed again in 3.7% formaldehyde in PBS for 10 min at room temperature. An additional fixation step was performed in 3.7% formaldehyde prepared in HEPES-buffered PBS (pH 7.4) for 60 min at room temperature. To improve probe penetration, nematodes were further digested with proteinase K (20 µg/mL) for 30 min at room temperature. To reduce non-specific probe binding, samples were treated with 0.1% triethanolamine (Sigma-Aldrich Korea) for 2 min, followed by incubation in 0.05% acetic anhydride in triethanolamine for 10 min. Hybridization was performed in hybridization buffer containing 50% formamide, 5× SSC, 0.1% Tween-20, 100 µg/mL heparin, and 100 µg/mL tRNA at 65 °C for 18 h with gentle rotation. A total of 1 µL of FAM-labeled antisense or sense probe (10 pmol) was used for each reaction. Post-hybridization washes were conducted using 2× SSC with 0.1% Tween-20 at room temperature, followed by a stringent wash in 0.2× SSC at 65 °C for 30 min and a final rinse in PBS. Imaging was carried out using an inverted fluorescence microscope (Eclipse 80i, Nikon, Tokyo, Japan) at 100× magnification using appropriate filters for FAM fluorescence detection.

### 2.11. Preparation of Chitosan-Formulated dsRNA

Chitosan formulation followed the method described in our previous study [25]. Chitosan (3.8–20 kDa, ≥75% deacetylated; Sigma-Aldrich Korea), derived from crab shells, was dissolved in 0.1 M sodium acetate buffer (prepared by mixing 0.1 M sodium acetate and 0.1 M acetic acid in deionized water, pH 4.5) and maintained at room temperature. For nanoparticle formation, 50 µL of dsRNA (0.5 µg/µL) was mixed with 100 µL of 100 mM sodium sulfate solution (Na_2_SO_4_ dissolved in deionized water). This solution was added to 100 µL of the chitosan solution (1 µg/µL) and incubated at 55 °C for 1 min, followed by mixing for 30 s. The resulting suspension was centrifuged at 14,000× *g* for 10 min at room temperature. The pellet (~300 nm in diameter) was washed three times with deionized water and resuspended in Milli-Q ultrapure water (Merck KGaA, Darmstadt, Germany). Nanoparticles were sonicated for 5 min at 25 °C using an ultrasonic liquid processor (Powersonic 405, Hwashin, Shanghai, China) prior to use in subsequent experiments. The final chitosan formulation contained dsRNA at a concentration of 500 µg/mL.

### 2.12. Statistical Analysis

All analyses were performed using one-way ANOVA with the PROC GLM procedure in the SAS program version 9.45 [26]. Mortality data were subjected to arcsine transformation before ANOVA. Mean comparisons were conducted using the least significant difference (LSD) test. The study included three independent biological replicates, and results were expressed as the mean ± standard error, generated with GraphPad v8.1 and Sigma Plot v10.

## 3. Results

### 3.1. Identification of a Nematode Isolate Collected from an Agricultural Land

Female adult nematode isolate was identified with nine morphological characteristics at the mouth, pharynx, esophagus, anus, and tail (Figure 1). The morphometric characteristics of the isolate matched to those of *Acrobeloides nanus* (Table 1). To confirm the morphological identification, the ITS sequence of the isolate was sequenced (Appendix A). Blast analysis to the GenBank supported the morphological identification with the highest match score to *A. nanus* (Table 2). A phylogenetic analysis indicated that the isolate is classified into one of the family Cephalobidae, in which it is clustered with *A. nanus* (Figure 2).

### 3.2. Prediction of Three Target Genes of RNAi from A. nanus Genome

Two structural genes, *Pat-10* and *Unc-87*, are essential for nematode survival and movement and their RNAi led to significant mortalities in *C. elegans* and plant parasitic nematode [27]. In addition, *vATPase-B* is a gene encoding a catalytic subunit (Figure 3) of the enzyme, which actively transports proton to the lumen of the midgut for the digested nutrients to be absorbed and is crucial for survival for several insects [25,28]. These three genes were retrieved from *A. nanus* genome and confirmed by Blast search to the GenBank (Table 3).

Their expressions were confirmed from the growing nematodes of *A. nanus* by RT-qPCR in two different developmental stages (Figure 4). Expression levels of the three genes were not different at the juvenile stage but they were at the adult stage, at which *Pat-10* was highly expressed more than other two genes.

### 3.3. Nematicidal Effects of Three dsRNAs Against A. nanus Using an Agar Plate Assay

On the growing medium, the nematodes of *A. nanus* were exposed to dsRNAs (Appendix A) specific to three target genes (Figure 5). All three dsRNAs gave significant mortalities (*F* = 30.87; df = 2, 24; *p* < 0.0001) to the nematodes in a dose-dependent manner (*F* = 359.67; df = 3, 24; *p* < 0.0001) (Figure 5a). However, there was a significant difference (*F* = 8.56; df = 6, 24; *p* < 0.0001) among three dsRNA treatments, in which dsRNA specific to *vATPase-B* showed the highest nematicidal effect with the lowest LC_50_ value (Appendix A). To confirm the mortality caused by the reduction in the target genes, the target genes were monitored in their amounts in the treated nematodes (Figure 5b). All three dsRNA treatments led to over 50% reduction in the expression levels of their target genes at 24–48 h after the dsRNA treatments.

To visualize the mRNA reduction caused by the dsRNA treatment, a fluorescence probe specific to *vATPase-B* mRNA was applied to the test nematodes (Figure 6). This FISH analysis showed that this gene was highly expressed in the gut in control (Figure 6a). The mRNA signal was markedly reduced in the nematode treated with dsRNA specific to *vATPase-B*. Based on the fluorescence signal intensity, the dsRNA treatment reduced the mRNA levels by more than 12-fold (Figure 6b).

### 3.4. Nematicidal Effects of Three dsRNAs Against A. nanus in Soil

To be practical for the dsRNA control against the soil nematode, the infective juveniles were applied to soil along with dsRNA (Figure 7). After 7 days, the nematodes in the treated soil were collected and assessed in the control efficacies of the three different dsRNA treatments. Unlike in the microplate assay, the dsRNAs specific to Unc-87 or Pat-10 gave low control efficacies below 40% and did not exhibit significant dose-dependent mortality against *A. nanus* (Figure 7a). In contrast, the dsRNA treatment specific to *vATPase-B* gave significant dose-dependent mortalities to the nematodes, in which dsRNA specific to *vATPase-B* achieved approximately 57% control efficacy at 500 µg/g (Figure 7b).

### 3.5. Chitosan-Formulated dsRNA to Protect dsRNA Stability in Soil

To increase the control efficacy of the dsRNAs in the soil by enhancing dsRNA stability, chitosan was used to formulate the dsRNAs (Figure 8). After the formulated dsRNA was first applied to soil, the test nematodes were periodically released to the treated soil. In unformulated dsRNA treatments, the control efficacies significantly decreased with the time in the soil before the nematode treatment (Figure 8a). Furthermore, they did not show any nematicidal activities in any treatments when the nematodes were exposed to dsRNA already sprayed 7 days previously in the soil. Interestingly, the formulated dsRNA did not exhibit a decline in nematicidal activity against *A. nanus*. The formulated dsRNA specific to *vATPase-B* significantly enhanced the nematicidal activity, achieving approximately 80% mortality against *A. nanus* (Figure 8b).

## 4. Discussion

This study tested the hypothesis that nematodes can be controlled via environmental application of dsRNA. A model nematode for this study used *A. nanus*, which were identified by morphological and molecular characteristics. Even though *A. nanus* is not a pest but rather a decomposer in the ecosystem, it is an ideal nematode species for developing a novel control agent using RNAi due to its well-known gene expression system [29,30]. In particular, its habitat in soil allowed us to test dsRNA application in soil using a chitosan formulation. This would be necessary to develop dsRNA to kill nematodes, which cause serious economic damage to crops because several nematodes such as root-knot nematodes, root lesion nematodes, or cyst nematodes live in soil environments. The isolate used in this study was collected from an agricultural farm cultivating cucumber. In contrast, the first report of this species in Korea was isolated in the soil collected from a potato farm [31]. This explains its habitats in a wide range of terrestrial areas, including forests, sand dunes, and agricultural farms [32].

Three different genes of *A. nanus* were silenced by treating specific dsRNAs to the living habitats. In a Petri dish, the dsRNA sprayed onto the growth medium significantly reduced the target gene expression and led to significant mortalities. Several nematode genes have been determined as effective targets of RNAi for pest control. Dong et al. [33] tested two kinds of genes of FLPs (FMRFamide-like peptides: modulating locomotory, feeding, and reproductive functions of nematodes) and mitogen-activated protein (MAP) kinase (regulating transcription factors and protein kinases) to control *Meloidogyne incognita*. In this study, test nematodes soaked in the dsRNA suspension exhibited a significant reduction in the target genes and suffered from poor infectivity of host plants by reducing root-knot numbers along with significant fecundity reduction in producing egg masses. Another two genes, *Unc-87* and *Pat-10*, of *C. elegans* are essential components of the body wall muscle and are thus required for nematode movement. Joseph et al. [34] used these genes as RNAi targets to control a banana root nematode, *Pratylenchus coffeae*. The nematode soaking treatment of these dsRNAs resulted in significant reductions in the mRNA levels in a sequence-specific manner and led to behavioral abnormalities, with the nematodes exhibiting a straight and rigid form in *Pat-10* RNAi treatment, while they exhibited coiled behavior in *Unc-87* RNAi treatment, in contrast to the regular sinusoidal movement of the control nematodes. Another structural gene, *Unc-15*, was an effective RNAi target for the control of a stem nematode, *Ditylenchus destructor*, which is one of the most serious diseases limiting the productivity and quality of sweet potato [35]. Two cuticle collagen genes, Mi-col-1 and Lemmi-5, of *M. incognita* are involved in the synthesis and maintenance of the nematode cuticle and highly expressed in adult females. Transgenic tomatoes expressing these dsRNAs reduced the number of the adult females, which critically impaired the nematode fecundity [36]. In addition, two transcriptional factors, *DAF-16* and *SKN-1*, and two nematode effectors, Mi-msp10 and Mi-msp23, are RNAi targets to control *M. incognita* [37,38]. Thorat et al. [39] tested nematode-responsive promoter of *A. thaliana* in tomato in order to express the dsRNA specifically upon the nematode damage and showed pAt1g74770 as a persistently inducible promoter during the nematode infection.

Among three candidate genes, *vATPase-B* was highly effective for controlling the nematode, *A. nanus*, in microplate and soil assays. However, the other two target genes were less potent due to their poor lethality to the nematode. In general, at least ~40% of genes are essential for maintaining life in most organisms [40]. Thus, any interruption of the expression of the essential genes leads to fatal lethality. However, the lethal effects after RNAi vary among these essential genes, probably due to the presence of paralogs or overexpression of alternative gene(s) [41]. For vATPase, its proton pumping function is crucial for nutrient absorption in the midgut in insects and nematodes [42,43]. Our current FISH assay showed that *vATPase-B* was highly expressed in the nematode gut of *A. nanus*. The mortality induced by the reduction in *vATPase-B* mRNA level has been reported in several insects and *C. elegans* [44,45].

This study developed a technique to apply dsRNA for controlling nematodes in soil. For this application, this study applied a chitosan formulation of dsRNA to enhance its chemical stability in the soil environment. In general, the half-life of dsRNA in soil is approximately 24 h due to various environmental factors including microbial degradation or absorption to soil particles [46]. The stabilized chitosan formulation significantly enhanced the nematicidal activity of the dsRNA. Chitosan formulation activated clathrin-dependent endocytosis pathway to enhance uptake efficiency of dsRNA by inducing the gene expression of the key clathrin heavy chain, which led to a significant increase in RNAi efficiency [47]. Chitosan formulation is also helpful to minimize its cellular degradation in the target cells by evacuating the dsRNA–chitosan from lysosomal endosomes [48]. A similar dsRNA formulation using a kaolinite nanoclay was devised in controlling a soil nematode, *G. pallida*, infesting potato crop [15]. Its drenching application led to significant impairment of the nematode juveniles, suggesting a potential to control the nematodes using dsRNA. Although little is known about effective dsRNA delivery methods to control nematodes in soil, this study suggests the chitosan formulation in addition to kaolinite nanoclay as a promising method. Even though our bioassay showed the nematicidal effect on the nematodes in soil, the chitosan-formulated dsRNA should be assessed for control efficacy against nematodes within plant roots because a number of the plant-parasitic nematodes reside in the plant tissues. In addition, because the amount of dsRNA required may be high to expect a successful control efficacy, this nematode control must be economically practical by screening highly efficient target genes to increase the RNAi efficiency and lead to high mortality. Because RNAi is sequence-specific, dsRNA-based nematode control must be target-specific. The optimal dsRNA should minimize off-target risks by selecting highly specific regions within effective target genes.

## 5. Conclusions

This study suggests a novel control tactic of soil-dwelling nematodes by drenching with a formulated dsRNA. It also provides a model nematode, *A. nanus*, to screen candidate genes to select highly nematicidal dsRNAs. The resulting dsRNA may be applicable for developing a highly efficient control method against soil nematode pests damaging crops.

## Figures and Tables

**Figure 1 biology-14-01161-f001:**
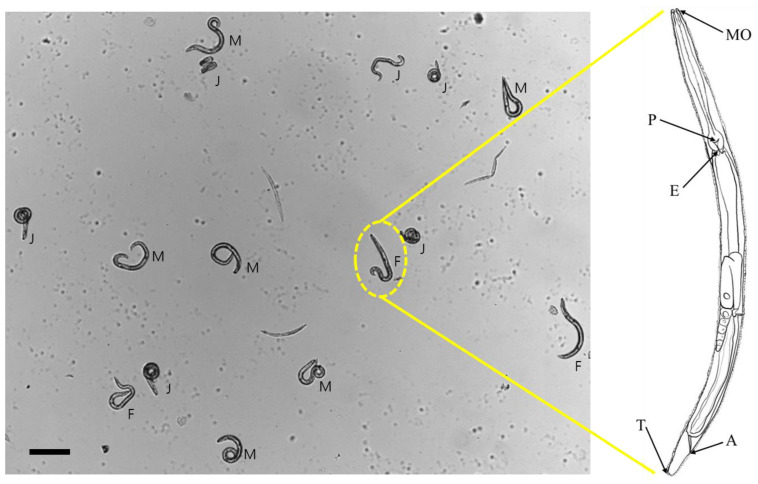
Nematode identification using morphological characteristics. Anatomical illustration of a nematode, highlighting key structures: mouth (‘MO’), esophagus (‘E’), pharynx (‘P’), tail (‘T’), and anus (‘A’). Juveniles (‘J’) are distinct from reproductive adults, which have ovary for females (‘F’) and testis for males (‘M’). Scale bar represents 100 μm.

**Figure 2 biology-14-01161-f002:**
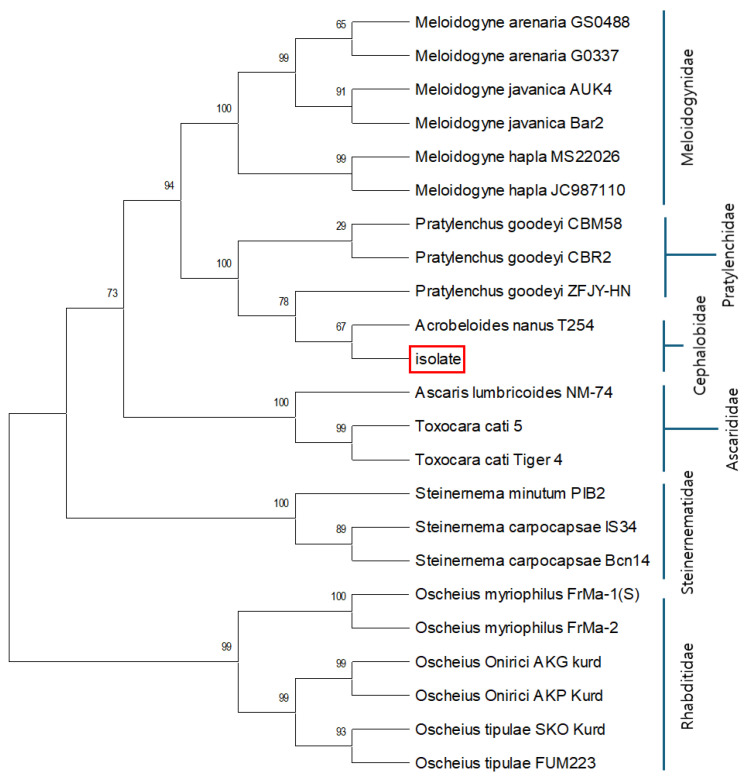
Phylogenetic analysis of the nematode isolate (red-boxed) based on ITS sequence. The phylogenetic tree was constructed using the Neighbor-Joining method in MEGA6.0, with bootstrap values calculated from 1000 replicates to support branch confidence. Sequences were retrieved from GenBank, with accession numbers provided in the corresponding dataset (Appendix A).

**Figure 3 biology-14-01161-f003:**
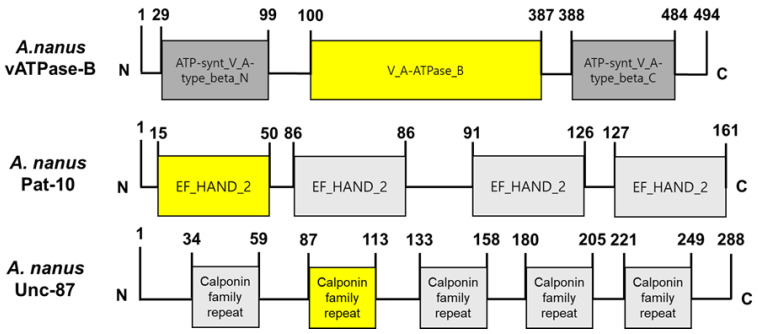
Three candidate genes of *A. nanus* for dsRNA construction. Domain analysis was conducted using InterPro (https://www.ebi.ac.uk/interpro/, accessed on 1 July 2025). Yellow colors indicate the signature domains of the specific genes.

**Figure 4 biology-14-01161-f004:**
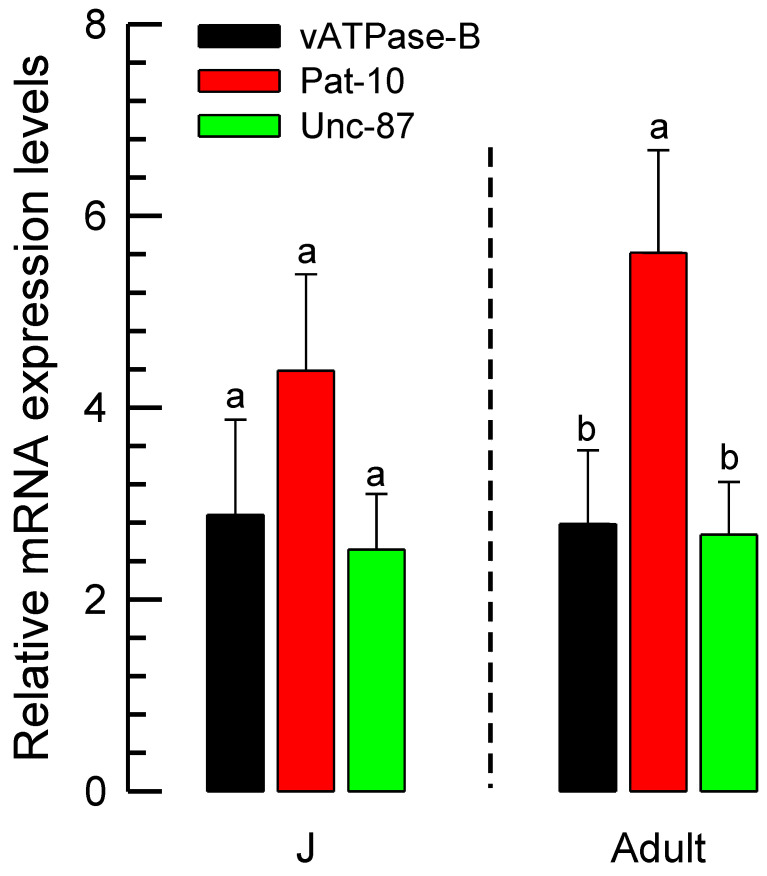
Expression profile of three genes at juvenile (‘J’) and adult stages of *A. nanus*. An-EFα1, an elongation factor, was employed to normalize the expression levels. Three replications were conducted for each measurement. The standard deviation bars with different letters indicate significant differences among means in each developmental stage at a Type I error = 0.05 (LSD test).

**Figure 5 biology-14-01161-f005:**
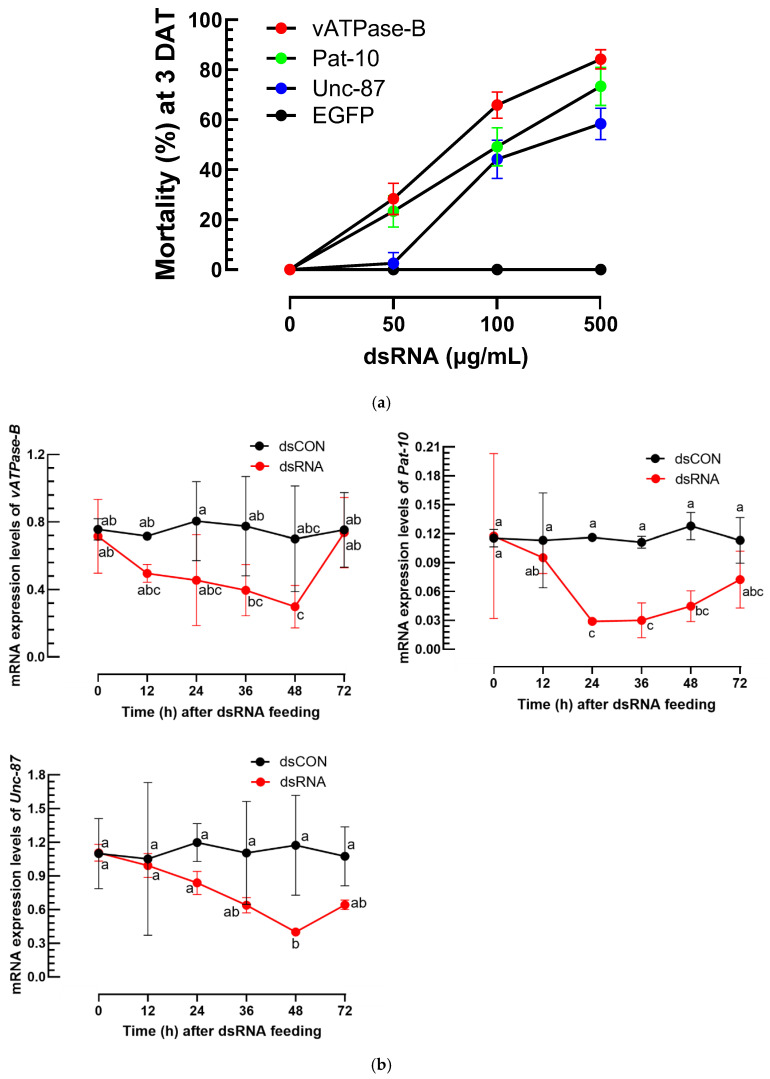
Efficacy of dsRNA application against *A. nanus* using an agar plate assay. (**a**) Nematode susceptibility to three different dsRNAs at different doses. *EGFP* was used as a control for a non-target gene. Mortality (%) was assessed 3 days after treatment (‘DAT’). Each plate contained 40 nematodes (a mixture of larvae and adults) and was replicated three times. Median lethal concentrations were estimated and shown in Appendix A. (**b**) Quantification of target gene expression levels after the RNAi treatments, where expression levels were normalized using an elongation factor, *EF1α*, for *A. nanus*. Control (‘dsCON’) utilized dsRNA specific to a green fluorescence protein gene, *EGFP*, serving as a non-target gene. Data represent mean ± standard deviation from three biological replicates. Different letters at the dots indicate significant difference among means at Type I error = 0.05 (LSD test).

**Figure 6 biology-14-01161-f006:**
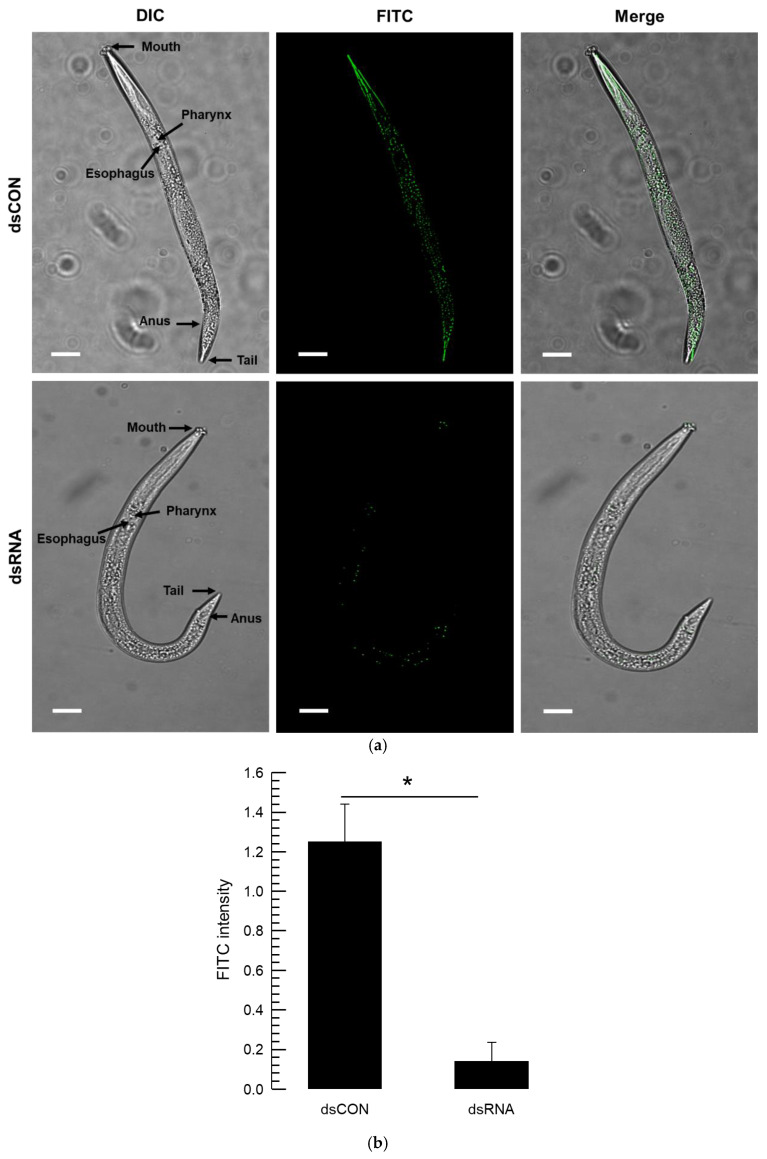
Comparative analysis of RNAi efficiency of a dsRNA treatment specific to *vATPase-B* in *A. nanus* using FISH. (**a**) Whole-mount in situ hybridization analysis in the nematode using antisense probes. Sense probes were employed to confirm specificity and did not show any signal. A fluorescent microscope (DM2500; Leica, Wetzlar, Germany) was used to examine the samples in fluorescence (‘FITC’ against *vATPase-B*). The overall nematode morphology was examined in differential interference contrast (‘DIC’) mode at 100× magnification, with a scale bar indicating 0.1 mm. (**b**) Fluorescence intensity was quantified by calculating the FITC signal. Each treatment was replicated three times, with each replication including three larvae. The asterisk above the standard deviation bars indicates the significant difference between means at Type I error = 0.05 (LSD test).

**Figure 7 biology-14-01161-f007:**
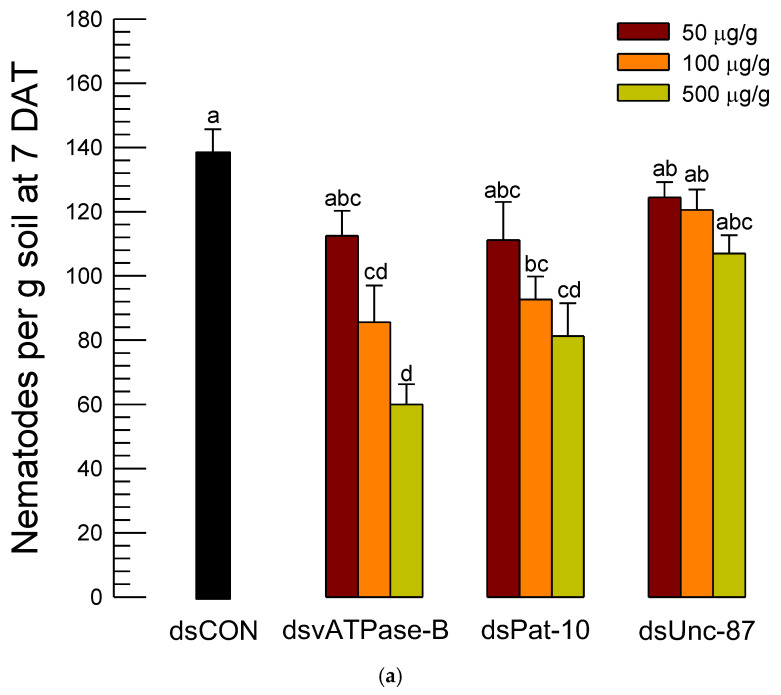
Efficacy of dsRNA application against *A. nanus* in soil. (**a**) Number of live nematodes at seven days after treatment (‘DAT’) in each dsRNA treatment: dsRNA specific to *vATPase-B* (‘dsvATPase-B’), dsRNA specific to *Pat-10* (‘dsPat-10’), and dsRNA specific to *Unc-87* (‘dsUnc-87’). Control (‘dsCON’) represented dsRNA treatment specific to a non-target gene, *EGFP*. Bars with different letters mean significant difference among means at Type I error = 0.05 (LSD test). (**b**) Control efficacies of the dsRNA treatments.

**Figure 8 biology-14-01161-f008:**
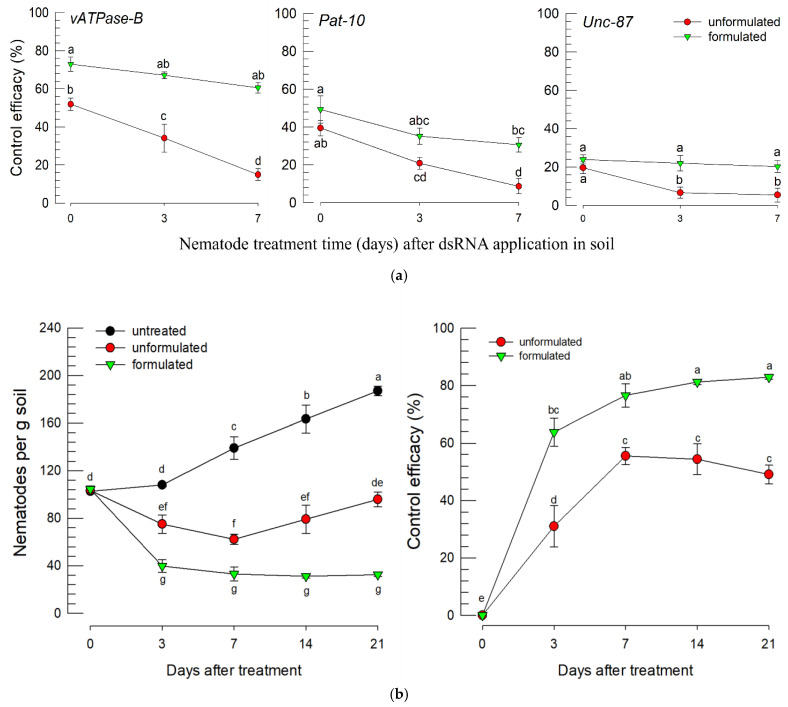
Influence of chitosan formulation of dsRNA on the control efficacy against *A. nanus*. (**a**) Comparison of dsRNA stability in soil conditions between formulated or unformulated dsRNAs. After incubation period of dsRNA in soil, the test nematodes (100 nematodes (a mixture of larvae and adults) per replicate) were applied to the pre-incubated soil–dsRNA mixture. Three dsRNA treatments (dsvATPase-B, dsPat-10, and dsUnc-87) were tested at 500 µg/g, while the control (dsCON) received dsRNA specific to EGFP. Each treatment was replicated three times. Different letters above the standard deviation bars indicate the significant difference between means at Type I error = 0.05 (LSD test). (**b**) Relatively longer control efficacy of chitosan-formulated dsRNA.

**Table 1 biology-14-01161-t001:** A morphometric comparison with characteristics of other nematode species (*Acrobeloides nanus, Pratylenchus goodeyi*, and *Meloidogyne incognita*). Homology percentages indicate morphological similarities between the isolate and the reference species.

Characters	Mean ± SD (95% CI)			
	Isolate	*A. nanus*	*P. goodeyi*	*M. incognita*
Body length (BL), µm	391.9 ± 28.8 (351.9~419.7)	408.6 ± 29.0 (335.3~442.3)	522.5 ± 15.9 (487~710)	417 ± 15 (387~459)
Body width (BW), µm	22.6 ± 2.8 (16.5~26.2)	21.3 ± 2.1 (17.1~24.5)	21.5 ± 4.8 (16.6~27.1)	21.8 ± 5.8 (15.8~27.8)
Tail length (T), µm	23.4 ± 3.4 (18.8~30.0)	23.4 ± 1.4 (20.5~25.4)	36.0 ± 5.0 (27.7~41)	49 ± 9.5 (36~56)
Anus width (AW), µm	12.9 ± 1.1 (10.7~14.2)	12.5 ± 1.4 (10.5~14.0)	7.8 ± 2.7 (5.0~10.9)	7.5 ± 0.8 (7.1~8.0)
BL/BW	17.3 ± 1.3 (15.4~19.5)	19.2 ± 1.1 (17.1~21.5)	24.3 ± 3.3 (19.7~29.5)	30.6 ± 4.1 (27.1~35.9)
BL/M to P	3.1 ± 0.4 (2.6~3.5)	3.3 ± 0.1 (3.0~3.5)	5.8 ± 0.9 (4.1~8.1)	2.24 ± 0.58 (2.10~3.35)
BL/M to E	3.4 ± 0.4 (2.8~4.1)	3.4 ± 0.4 (2.8~4.1)	4.9 ± 0.8 (2.9~5.3)	7.5 ± 0.5 (7.1~8.0)
BL/T	13.6 ± 2.0 (11.2~17.4)	17.5 ± 0.8 (15.8~19.0)	13.8 ± 0.3 (13.8~20.0)	8.5 ± 2.8 (7.3~11.1)
BL/AW	24.4 ± 1.4 (21.4~26.5)	33.03 ± 9.1 (24.0~42.1)	67.0 ± 4.0 (62.5~71.2)	55.6 ± 8.3 (46.9~65.8)
Homology to isolate (%)	100% (9/9)		44.4% (4/9)	33.3% (3/9)

**Table 2 biology-14-01161-t002:** Molecular identification and phylogenetic analysis of a nematode isolate Gunwi using its ITS sequence (GenBank accession number: PV981766). BLAST analysis of the ITS region of the nematode isolates against the GenBank database. The table lists the five top-matching species, accession numbers, match scores, query cover, E-value, and sequence identity.

Blasted Species	GenBank Accession Number	Match Score	Query Cover (%)	E-Value	Identity (%)
*Acrobeloides nanus*	KY828308.1	1354	92	0.00	99.5
*Pratylenchus goodeyi*	KM874803.1	1227	91	0.00	96.8
*Zeldia punctata*	ON738667.1	859	92	0.00	87.9
*Aphelenchoides arachidis*	EF371501.1	811	88	0.00	87.2
*Heterorhabditis indica*	MK271288.1	601	91	6.00 × 10^−167^	81.7

**Table 3 biology-14-01161-t003:** Bioinformatic analysis of three distinct genes of *A. nanus* obtained from the GenBank database. The BLAST tables list the five top-matching genes, species, accession numbers, match scores, query coverage, E-value, and sequence identity.

Gene	Species	GenBank Number	Match Score	Query Cover (%)	E-Value	Identity (%)
** Match to *A. nanus* vATPase-B **						
**vATPase-B**	*Heamonchus contortus*	WOA00646.1	1021	100	0	100
**vATPase-B**	*Parelaphostrongylus tenuis*	KAJ1373751.1	1003	99	0	97.96
**vATPase-B**	*Dictyocaulus viviparus*	KJH42266.1	986	100	0	91.49
**vATPase-B**	*Necator americanus*	ETN82885.1	976	100	0	94.53
**vATPase-B**	*Oesophagostomum dentatum*	KHJ93652.1	970	99	0	94.70
** Match to *A. nanus* Pat-10 **						
**Troponin C**	*Caenorhabditis brenneri*	EU677743.1	452	89	5.0 × 10^−75^	82.53
**Pat-10**	*Brugia malayi*	XM_001901190.2	315	99	2.0 × 10^−80^	72.34
**EF-hand**	*Onchocerca volvulus*	GQ202196.1	291	100	2.0 × 10^−73^	71.21
**Troponin C**	*Caenorhabditis elegans*	D45896.1	434	89	3.0 × 10^−72^	81.85
**Troponin**	*Anisakis simplex*	AJ012103.2	436	89	1.0 × 10^−69^	81.16
** Match to *A. nanus* Unc-87 **						
**Calponin**	*Pratylenchus coffeae*	JQ929656.1	424	66	5.0 × 10^−119^	88.22
**Calponin-1**	*Meloidogyne incognita*	AJ277868.1	579	95	5.0 × 10^−113^	78.87
**Unc-87**	*Heterodera glycines*	AY672636.1	571	99	4.0 × 10^−102^	76.67
**Unc-87**	*Caenorhabditis elegans*	NM_001025922.6	397	100	3.0 × 10^−78^	73.06
**Pat-10**	*Brugia malayi*	XM_001901190.2	240	92	1.0 × 10^−57^	71.04

## Data Availability

No new data were created or analyzed in this study.

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
