# Peer review of "Nematicidal Efficacy of a dsRNA-Chitosan Formulation Against Acrobeloides nanus Estimated by a Soil Drenching Application"

_biology, 2025, doi:10.3390/biology14091161_

Round 1

Reviewer 1 Report

Comments and Suggestions for Authors

Recommendation: Minor revision

This manuscript reports on the use of spray-induced gene silencing (SIGS) to control Acrobeloides 
nanus, a free-living nematode, by applying dsRNA targeting essential genes. It emphasizes the 
improvement of dsRNA stability and efficacy through chitosan-based formulation. The study is well
structured, the experiments are clearly described, and the use of both qPCR and FISH to verify gene 
knockdown adds rigor. The findings contribute meaningfully to the growing interest in RNA-based 
biocontrol alternatives to conventional nematicides. 
However, there are several areas that need revision before the manuscript can be accepted. In 
particular, the discussion section would benefit from a more critical assessment of SIGS practicality 
and its potential application beyond laboratory conditions. Additionally, there is a significant issue 
regarding gene sequence transparency. 
Gene Sequence Availability 
The manuscript explains that A. nanus gene targets were identified by BLAST alignment using 
orthologs from species like C. elegans and P. goodeyi. However, the actual A. nanus sequences used 
for RNAi and qPCR were not deposited in GenBank or included in the supplementary material. While 
primer sequences are provided in Table S1, the dsRNA fragments or amplicons themselves are not 
shown, and there is no clear evidence that these sequences are unique to A. nanus or validated 
beyond similarity-based prediction. 
Recommendation: The authors should be encouraged to share the exact sequences of the gene 
fragments used in their RNAi experiments, either as supplementary data or via public database 
submission. This will enhance transparency and allow reproducibility. 
Major issues to address: 
1. SIGS practicality in real soil conditions is not fully discussed 
The study shows that unformulated dsRNA rapidly loses activity in soil, but real field conditions are 
likely even more challenging due to microbial activity, UV exposure, and binding to soil particles. The 
authors should acknowledge that while chitosan improves stability, further improvements will be 
needed for field use. 
2. Not all dsRNA targets were effective in soil 
While SIGS targeting vATPase-B performed well, Pat-10 and Unc-87 did not. The discussion should 
reflect on possible reasons for this variation, such as dsRNA length, degradation rate, or differences 
in gene expression. This will help clarify that efficacy is gene- and context-dependent. 

3. Use of a model organism limits generalizability 
A. nanus is not a plant-parasitic nematode. Although it is useful as a model, the discussion and 
conclusion should be careful not to overextend the results. Claims regarding the control of 'serious 
soil nematodes' should be softened unless supported by data from pest species. 
4. Limited reach of external application in endoparasitic nematodes 
Most major nematode pests spend part of their lifecycle inside plant tissues. Soil drenching may not 
reach them after invasion. This limitation should be acknowledged, and possible improvements in 
timing or delivery strategies discussed. 
5. No discussion of dsRNA production cost and scalability 
The effective concentration used (500 ppm) is quite high. Scaling this up for field use would be 
expensive without efficient production systems. The authors should briefly mention this and 
reference current work on low-cost dsRNA production platforms (e.g., microbial or chloroplast 
expression). 
6. Off-target risks are not considered 
Since A. nanus is not a pest, the study itself shows that non-target species can be affected by 
environmental dsRNA. This raises important biosafety concerns. A sentence or two on species
specific design and risk assessment would be appropriate. 
7. Chitosan formulation is treated too simplistically 
While the use of chitosan clearly improves stability, its effectiveness depends on formulation 
properties such as molecular weight and deacetylation. These factors can affect uptake and 
consistency. The authors should briefly mention this and the need for standardization. 

Minor issues to correct: 
• Abstract: “lInc-87” should be corrected to “Unc-87” 
• Abstract: “straegy” should be corrected to “strategy” 
• Author section: “Correspondence: Correspondence:” is duplicated and should be fixed 
• Gene names (e.g., unc-87, pat-10) should be consistently formatted, ideally italicized 
• Units such as “ppm” should be clarified as µg/mL or µg/g 
• Figure 7 - In panel (a), “dsvATPase-B” is labeled on the X-axis, while panel (b) just uses 
“vATPase-B”. Harmonizing these across panels and with the figure caption would reduce any 
potential confusion. Suggested: use dsvATPase-B, dsPat-10, and dsUnc-87 consistently in 
both panels and the text. 
• The discussion should explicitly note that A. nanus is a model organism and not a plant pest. Overall, this is a solid and meaningful contribution to the field of RNAi-based nematode control. The authors demonstrate a clear proof of concept that chitosan-formulated dsRNA can be used to achieve nematicidal effects in a soil environment. However, to make this work suitable for 
publication, the authors should revise the discussion to more accurately reflect the current 
limitations of SIGS, address the missing gene sequence information, and make the small corrections noted above. 
Recommendation: Minor revision. 

Comments on the Quality of English Language

The manuscript is clearly written overall, and the scientific content is communicated effectively. However, a few minor typographical errors and occasional phrasing issues slightly affect the flow in some sections. With light editing and careful proofreading, particularly to ensure consistency in gene name formatting and to smooth a few sentences in the discussion, the language can be further improved. The clarity of writing already provides a strong foundation.

Author Response

Comment #1-1: This manuscript reports on the use of spray-induced gene silencing (SIGS) to control Acrobeloides nanus, a free-living nematode, by applying dsRNA targeting essential genes. It emphasizes the improvement of dsRNA stability and efficacy through chitosan-based formulation. The study is well structured, the experiments are clearly described, and the use of both qPCR and FISH to verify gene knockdown adds rigor. The findings contribute meaningfully to the growing interest in RNA-based biocontrol alternatives to conventional nematicides. However, there are several areas that need revision before the manuscript can be accepted. In particular, the discussion section would benefit from a more critical assessment of SIGS practicality and its potential application beyond laboratory conditions. Additionally, there is a significant issue regarding gene sequence transparency.

Response: We appreciate your understanding on our manuscript. The critical issues you raised are addressed by adding additional explanations in a point-by-point manner.

Comment #1-2: Gene Sequence Availability. The manuscript explains that A. nanus gene targets were identified by BLAST alignment using orthologs from species like C. elegans and P. goodeyi. However, the actual A. nanus sequences used for RNAi and qPCR were not deposited in GenBank or included in the supplementary material. While primer sequences are provided in Table S1, the dsRNA fragments or amplicons themselves are not shown, and there is no clear evidence that these sequences are unique to A. nanus or validated beyond similarity-based prediction.

Response: The COI sequence of the nematode was provided in Supplementary Information at the original submission. The sequence was submitted to GenBank to get the accession number.

Comment #1-3: Recommendation: The authors should be encouraged to share the exact sequences of the gene fragments used in their RNAi experiments, either as supplementary data or via public database submission. This will enhance transparency and allow reproducibility.

Response: The sequences used for dsRNAs were added to the Supplementary Information in Figure S2.

Comment #1-4: SIGS practicality in real soil conditions is not fully discussed. The study shows that unformulated dsRNA rapidly loses activity in soil, but real field conditions are likely even more challenging due to microbial activity, UV exposure, and binding to soil particles. The authors should acknowledge that while chitosan improves stability, further improvements will be needed for field use.

Response: The unstable dsRNA in soil is clearly added in the discussion as follows: “Dubelman, S., Fischer, J., Zapata, F., Huizinga, K., Jiang, C., Uffman, J., Levine, S., Carson, D., 2014. Environmental fate of double-stranded RNA in agricultural soils. PLoS ONE 9, e93155.” In addition to chitosan formulation, we suggest another nano-formulation using kaolinite.

Comment #1-5: Not all dsRNA targets were effective in soil. While SIGS targeting vATPase-B performed well, Pat-10 and Unc-87 did not. The discussion should reflect on possible reasons for this variation, such as dsRNA length, degradation rate, or differences in gene expression. This will help clarify that efficacy is gene- and context-dependent.

Response: We add the rationale as follows in the discussion: “Among three candidate genes, vATPase-B was highly effective to control the nema-tode, A. nanus, in microplate and soil assays. However, the other two target genes were less potent due to their poor lethality to the nematode. In general, at least 40% among the total genes are essential for maintaining life in most organisms [40]. Thus, any interruption of the expression of the essential genes leads to fatal lethality. However, the lethal effects after RNAi vary among these essential genes probably due to the present of paralogs or overex-pression of alternative gene(s) [41].”

Comment #1-6: Use of a model organism limits generalizability. A. nanus is not a plant-parasitic nematode. Although it is useful as a model, the discussion and conclusion should be careful not to overextend the results. Claims regarding the control of 'serious soil nematodes' should be softened unless supported by data from pest species.

Response: This comment is highly logical. We rephrased as follows: “The resulting dsRNA may be applicable for developing a highly efficient control method against soil nematode pests damaging crops.”

Comment #1-7: Limited reach of external application in endoparasitic nematodes. Most major nematode pests spend part of their lifecycle inside plant tissues. Soil drenching may not reach them after invasion. This limitation should be acknowledged, and possible improvements in timing or delivery strategies discussed.

Response: This is another excellent comment. We add the point to the discussion as follows: “Even though our bioassay showed the nematicidal effect on the nematodes in soil, the chitosan-formulated dsRNA needs to be assessed in its control efficacy against nematodes in the plant roots because a number of the plant-parasitic nematodes reside in the plant tissues.”

Comment #1-8: No discussion of dsRNA production cost and scalability. The effective concentration used (500 ppm) is quite high. Scaling this up for field use would be expensive without efficient production systems. The authors should briefly mention this and reference current work on low-cost dsRNA production platforms (e.g., microbial or chloroplast expression).

Response: We add this information to the discussion as follows: “In addition, due to too much amount of dsRNA to expect a successful control efficacy, this nematode control needs to be practical in the economic sense by screening highly efficient target gene to increase the RNAi efficiency and lead to high mortality.”

Comment #1-9: Off-target risks are not considered. Since A. nanus is not a pest, the study itself shows that non-target species can be affected by environmental dsRNA. This raises important biosafety concerns. A sentence or two on species specific design and risk assessment would be appropriate.

Response: We add this point to the discussion as follows: “While the nematode control using dsRNA must be target-specific because the RNAi is sequence-specific, the optimal dsRNA should avoid off-target risks by determining the highly specific region for dsRNA within the highly efficient target gene.”

Comment #1-10: Chitosan formulation is treated too simplistically. While the use of chitosan clearly improves stability, its effectiveness depends on formulation properties such as molecular weight and deacetylation. These factors can affect uptake and consistency. The authors should briefly mention this and the need for standardization.

Response: The concentrations of dsRNA and chitosan are added to the M&M.

Comment #1-11: Abstract: “lInc-87” should be corrected to “Unc-87”

Response: My version was OK.

Comment #1-12: Abstract: “straegy” should be corrected to “strategy”

Response: My version was OK.

Comment #1-13: Author section: “Correspondence: Correspondence:” is duplicated and should be fixed

Response: My version was OK.

Comment #1-14: Gene names (e.g., unc-87, pat-10) should be consistently formatted, ideally italicized

Response: All are italicized.

Comment #1-15: Units such as “ppm” should be clarified as µg/mL or µg/g

Response: All are corrected into ug/mL or ug/g.

Comment #1-16: Figure 7 - In panel (a), “dsvATPase-B” is labeled on the X-axis, while panel (b) just uses “vATPase-B”. Harmonizing these across panels and with the figure caption would reduce any potential confusion. Suggested: use dsvATPase-B, dsPat-10, and dsUnc-87 consistently in both panels and the text.

Response: All are consistent.

Comment #1-17: The discussion should explicitly note that A. nanus is a model organism and not a plant pest. Overall, this is a solid and meaningful contribution to the field of RNAi-based nematode control. The authors demonstrate a clear proof of concept that chitosan-formulated dsRNA can be used to achieve nematicidal effects in a soil environment. However, to make this work suitable for publication, the authors should revise the discussion to more accurately reflect the current limitations of SIGS, address the missing gene sequence information, and make the small corrections noted above.

Response: The additional issues are explained in the discussion and also state the conclusion as follows:

Even though our bioassay showed the nematicidal effect on the nematodes in soil, the chitosan-formulated dsRNA needs to be assessed in its control efficacy against nematodes in the plant roots because a number of the plant-parasitic nematodes reside in the plant tissues. In addition, due to too much amount of dsRNA to expect a successful control effi-cacy, this nematode control needs to be practical in the economic sense by screening highly efficient target gene to increase the RNAi efficiency and lead to high mortality. While the nematode control using dsRNA must be target-specific because the RNAi is se-quence-specific, the optimal dsRNA should avoid off-target risks by determining the highly specific region for dsRNA within the highly efficient target gene.

  1. Conclusion

This study suggests a novel control tactics of soil-dwelling nematodes by drenching a formulated dsRNA. It also provides a model nematode, A. nanus, to screen candidate genes for selecting highly nematicidal dsRNAs. The resulting dsRNA may be applicable for de-veloping a highly efficient control method against soil nematode pests damaging crops.

Reviewer 2 Report

Comments and Suggestions for Authors

This article developed chitosan-formulated dsRNA to control nematode, Acrobeloides nanus. The formulation has potent nematicidal efficacy and lower environment risk. The article has certain novelty, and can be considered for acceptance after minor revision.

  1. There are many grammatical errors in abstract

Line 14 ("morphology icaland molecular markers"):

There is a space missing between "morphology" and "ical". It should be "morphological and molecular markers."

Line 15 ("Spray-induced gene silencing using double-stranded RNA(dsRNA) represents a promising new strategy for pest control."):

There is a missing space between "RNA" and "(dsRNA)". It should be "RNA (dsRNA)".

Line 17 ("environmental spray of dsRNA that specifically suppressed target genes in A. nanus."):

The phrase "environmental spray of dsRNA" could be clearer. Consider rephrasing to:"spray of dsRNA for environmental application that specifically targets genes in A. nanus."

Line 28 ("straegy"):

"straegy" should be "strategy." “A. nanus”should be italicized.

Line 30:“Acrobeloides nanus”should't be bold.

Line 33:“Acrobeloides nanus”should be italicized.

  1. “The three genes were not different in their expression levels at juvenile stage but not at adult stage...”. This sentence may cause misunderstandings, please revise it.
  2. How to confirm that dsRNA and Chitosan have combined to form nanocomplexes?
  3. L35-36:“RNA interference (RNAi) is a... post-tran-scriptional leve.”This sentence is inaccurate.
  4. L65-73:“In this study, Globodera pallida, a white potato cyst nematode, is a target nematode,... we used a soil nematode, Acrobeloides nanus,”The description of the two nematodes here is confusing.
  5. L169:“dsCON”need a detailed explanation.
  6. L170-171:“40 individuals were transferred into...”Only 40 nematodes?
  7. L213-215:What is the diameter of the pellet?
  8. L227-278:Please separate the Figures and Tables.
  9. In fig.5, which developmental stage is used as the control for the expression levels of these three genes in larvae and adults? Please specify
  10. In fig.5, the mortality of Acrobeloides nanus after being treated with dsCON should be as the control to assess the control efficacy of dsRNA (dsPat-10, dslInc-87, and dsvATPase-B) application.
  11. Line 396: "used these genes as RNAi targets of a banana root nematode". This sentence structure is slightly unclear.
  12. Why did dsPat-10 show high mortality in agar (Fig. 5a) but minimal efficacy in soil (Fig. 7), while dsUnc-87 was ineffective in both? Does this suggest target-/tissue-specific barriers to RNAi delivery in soil?*
  13. Line 402: "effective target of the RNAi control of a stem nematode"Error: The phrasing is awkward. It should be "effective RNAi target for controlling a stem nematode.

Author Response

Comment #2-1: This article developed chitosan-formulated dsRNA to control nematode, Acrobeloides nanus. The formulation has potent nematicidal efficacy and lower environment risk. The article has certain novelty, and can be considered for acceptance after minor revision.

Response: We appreciate your full understanding of our manuscript.

Comment #2-2: There are many grammatical errors in abstract

Line 14 ("morphology icaland molecular markers"):

There is a space missing between "morphology" and "ical". It should be "morphological and molecular markers."

Response: My version was OK.

Comment #2-3: Line 15 ("Spray-induced gene silencing using double-stranded RNA(dsRNA) represents a promising new strategy for pest control."): There is a missing space between "RNA" and "(dsRNA)". It should be "RNA (dsRNA)".

Response: My version was OK.

Comment #2-4: Line 17 ("environmental spray of dsRNA that specifically suppressed target genes in A. nanus."): The phrase "environmental spray of dsRNA" could be clearer. Consider rephrasing to:"spray of dsRNA for environmental application that specifically targets genes in A. nanus."

Response: It is rephrased as follows: “Here, we tested a spraying dsRNA would specifically suppress the target genes in A. nanus.”

Comment #2-5: Line 28 ("straegy"): "straegy" should be "strategy." “A. nanus” should be italicized.

Response: My version was OK.

Comment #2-6: Line 30:“Acrobeloides nanus”should't be bold.

Response: My version was OK.

Comment #2-7: Line 33:“Acrobeloides nanus”should be italicized.

Response: My version was OK.

Comment #2-8: “The three genes were not different in their expression levels at juvenile stage but not at adult stage...”. This sentence may cause misunderstandings, please revise it.

Response: My version was OK.

Comment #2-9: How to confirm that dsRNA and Chitosan have combined to form nanocomplexes?

Response: Based on our previous study (Khan et al., 2024), the optimal concentrations of chitosan and dsRNA were determined. Details are added to the M&M. Actual size of the nanoparticles are measured and used for preparing another manuscript, which is now under review. Thus, I wrote ~300 nm.

**** Here is a supporting figure ***** 

Comment #2-10: L35-36:“RNA interference (RNAi) is a... post-tran-scriptional leve.”This sentence is inaccurate.

Response: Rephrased as follows: “at a posttranscriptional level [1].”

Comment #2-11: L65-73:“In this study, Globodera pallida, a white potato cyst nematode, is a target nematode,... we used a soil nematode, Acrobeloides nanus,” The description of the two nematodes here is confusing.

Response: Rephrased as follows: “, in which Globodera pallida, a white potato cyst nematode, is a target nematode and is a major economic pest that causes substantial potato yield losses [15].”

Comment #2-12: L169:“dsCON” need a detailed explanation.

Response: Rephased as follows: “the control dsRNA (dsCON)”

Comment #2-13: L170-171:“40 individuals were transferred into...”Only 40 nematodes?

Response: For agar plate assay, each experimental unit was 40 nematodes. For soil test, it used 100 nematodes.

Comment #2-14: L213-215:What is the diameter of the pellet?

Response: ~300 nm (see the response to comment #2-9)

Comment #2-15: L227-278:Please separate the Figures and Tables.

Response: The figure indicates the morphological characters in the table. Thus, these two are in a figure.

Comment #2-16: In fig.5, which developmental stage is used as the control for the expression levels of these three genes in larvae and adults? Please specify

Response: Mixed stages of larvae and adults were used. This is added to the caption.

Comment #2-17: In fig.5, the mortality of Acrobeloides nanus after being treated with dsCON should be as the control to assess the control efficacy of dsRNA (dsPat-10, dslInc-87, and dsvATPase-B) application.

Response: Added to the revised figure.

Comment #2-18: Line 396: "used these genes as RNAi targets of a banana root nematode". This sentence structure is slightly unclear.

Response: Rephrased as follows: “used these genes as RNAi targets to control a banana root nematode”

Comment #2-19: Why did dsPat-10 show high mortality in agar (Fig. 5a) but minimal efficacy in soil (Fig. 7), while dsUnc-87 was ineffective in both? Does this suggest target-/tissue-specific barriers to RNAi delivery in soil?*

Response: This kind of differential RNAi efficacy is explained in the discussion as follows: “Among three candidate genes, vATPase-B was highly effective to control the nema-tode, A. nanus, in microplate and soil assays. However, the other two target genes were less potent due to their poor lethality to the nematode. In general, at least 40% among the total genes are essential for maintaining life in most organisms [40]. Thus, any interruption of the expression of the essential genes leads to fatal lethality. However, the lethal effects after RNAi vary among these essential genes probably due to the present of paralogs or overex-pression of alternative gene(s) [41].”

Comment #2-20: Line 402: "effective target of the RNAi control of a stem nematode" Error: The phrasing is awkward. It should be "effective RNAi target for controlling a stem nematode.

Response: Rephrased as follows: “was an effective RNAi target for controlling a stem nematode”

Reviewer 3 Report

Comments and Suggestions for Authors

The MS needs major revision in the writing style, grammar, use of phrases, and the figures.

  1. The MS is poorly written. The grammar may be checked carefully and rewritten accordingly.
  2. The use of ‘unnecessary articles’ in the MS may be checked carefully and removed.
  3. The MS needs major changes in the writing style and suggests improvement of various grammatical errors.
  4. Figures may be placed properly with the caption and grouped accordingly.
  5. Related figures may be clubbed together under a single frame.
  6. The author may highlight previous works on chitosan NP as well as dsRNA-chitosan works in the review and discussion section.
  7. What is the fate of the dsRNA in the soil?
  8. The author needs to check the font size in the MS and the figures, and should try to use a uniform font size.

Comments on the Quality of English Language

The MS is poorly written. The grammar may be checked carefully and rewritten accordingly.

Author Response

Comment #3-1: The MS is poorly written. The grammar may be checked carefully and rewritten accordingly. The MS needs major changes in the writing style and suggests improvement of various grammatical errors.

Response: All the typos and awkward fonts might be arisen during the conversion of the manuscript to the journal format. The original manuscript had little problem. In addition, the entire manuscript has been corrected with the helpful comments raised by the two other reviewers.

Comment #3-2: The use of ‘unnecessary articles’ in the MS may be checked carefully and removed.

Response: To minimize the unnecessary references, the revised manuscript has been carefully reviewed by the corresponding author including the additional references in the course of revision according to comments raised by two other reviewers.

Comment #3-3: Figures may be placed properly with the caption and grouped accordingly.

Related figures may be clubbed together under a single frame.

Response: All the figure captions have been reviewed by the corresponding author after revision. And each of the figures has been confirmed that it delivers a scientific message.

Comment #3-4: The author may highlight previous works on chitosan NP as well as dsRNA-chitosan works in the review and discussion section.

Response: This part in the discussion has been further supported by the additional explanations as follows: “This study developed a technique to apply dsRNA to control nematodes in soil. For this application, this study applied chitosan formulation of dsRNA to enhance its chemical stability in the soil environment. In general, the half-life of dsRNA in soil is roughly 24 h due to various environmental factors including microbial degradation or absorption to soil particles [46]. The stabilized chitosan formulation significantly en-hanced the nematicidal activity of the dsRNA. Chitosan formulation activated clath-rin-dependent endocytosis pathway to enhance uptake efficiency of dsRNA by induc-ing the gene expression of the key clathrin heavy chain, which led to significant in-crease of RNAi efficiency [47]. Chitosan formulation is also helpful to minimize its cellular degradation in the target cells by escaping the dsRNA-chitosan from lysosomal endosomes [48]. A similar dsRNA formulation using a kaolinite nanoclay was devised in controlling a soil nematode, G. pallida, infesting potato crop [15]. Its drenching ap-plication led to significantly impair the nematode juveniles, suggesting a potential to control the nematodes using dsRNA. Although little has been known in the effective dsRNA delivery methods to control nematodes in soil, this study suggests the chitosan formulation in additional to kaolinite nanoclay as a promising method. Even though our bioassay showed the nematicidal effect on the nematodes in soil, the chi-tosan-formulated dsRNA needs to be assessed in its control efficacy against nematodes in the plant roots because a number of the plant-parasitic nematodes reside in the plant tissues. In addition, due to too much amount of dsRNA to expect a successful control efficacy, this nematode control needs to be practical in the economic sense by screening highly efficient target gene to increase the RNAi efficiency and lead to high mortality. While the nematode control using dsRNA must be target-specific because the RNAi is sequence-specific, the optimal dsRNA should avoid off-target risks by de-termining the highly specific region for dsRNA within the highly efficient target gene.”

Comment #3-5: What is the fate of the dsRNA in the soil?

Response: The fate in the soil is explained with the following sentence: “In general, the half-life of dsRNA in soil is roughly 24 h due to various environmental factors including microbial degradation or absorption to soil particles [46].”

Comment #3-6: The author needs to check the font size in the MS and the figures, and should try to use a uniform font size.

Response: To be uniform, the entire manuscript has been reviewed and confirmed by the corresponding author.

Round 2

Reviewer 3 Report

Comments and Suggestions for Authors

Still many corrections are not been addressed even the title should be modified

Comments on the Quality of English Language

Still many corrections are not been addressed even the title should be modified

Author Response

All the comments in the text are reflected in the revised version. See the red-colored text in the attach file.

These include the title change and additional explanations on the subtitles.

Some typos were caused by the conversion from the original to the journal format and confirmed in a right way.

Hope this revision would be satisfied by the reviewer.

Round 3

Reviewer 3 Report

Comments and Suggestions for Authors

There are some change still need to be addressed in MS (highlited in the MS). Change the title to and more appropriate.

Author Response

Comment: There are some change still need to be addressed in MS (highlited in the MS). Change the title to and more appropriate.

Response: All comments are reflected in the revised version 3. Please see the red-colored text in biology-3995459-r3-main+SI-track.pdf file.
